# Multi-Agent Evolutionary Game in the Recycling Utilization of Sulfate-Rich Wastewater

**DOI:** 10.3390/ijerph19148770

**Published:** 2022-07-19

**Authors:** Meng Ding, Hui Zeng

**Affiliations:** 1School of Urban Planning and Design, Shenzhen Graduate School, Peking University, Shenzhen 518055, China; zengh@pkusz.edu.cn; 2IER Environmental Protection Engineering Technology Co., Ltd., Shenzhen 518071, China

**Keywords:** sulfate-rich wastewater (SRW), resources recovery, wastewater management, tripartite game, evolutionary game theory, environmental supervision

## Abstract

Current industrial development has led to an increase in sulfate-rich industrial sewage, threatening industrial ecology and the environment. Incorrectly treating high-concentration sulfate wastewater can cause serious environmental problems and even harm human health. Water with high sulfate levels can be treated as a resource and treated harmlessly to meet the needs of the circular economy. Today, governments worldwide are working hard to encourage the safe disposal and reuse of industrial salt-rich wastewater by recycling sulfate-rich wastewater (SRW) resources. However, the conflict of interests between the SRW production department, the SRW recycling department, and the governments often make it challenging to effectively manage sulfate-rich wastewater resources. This study aims to use the mechanism of evolutionary game theory (EGT) to conduct theoretical modelling and simulation analysis on the interaction of the behaviour of the above three participants. This paper focuses on the impact of government intervention and the ecological behaviour of wastewater producers on the behavioural decisions of recyclers. The results suggest that the government should play a leading role in developing the SRW resource recovery industry. SRW producers protect the environment in the mature stage, and recyclers actively collect and recover compliant sulfate wastewater resources. Governments should gradually deregulate and eventually withdraw from the market. Qualified recyclers and environmentally friendly wastewater producers can benefit from a mature SRW resources recovery industry.

## 1. Introduction

Rapid industrialization has led to a significant increase in demand for water for industry. Due to the limitation of water resources, experts and scholars are prompted to consider industrial wastewater’s sustainability and resource utilization. There is much sulfate in industrial wastewater, which comes from food production, medicines, printing, and coal mines [1]. Such sulfate-rich wastewater (SRW) will cause acidification of the water body, and it will also compact soil and negatively affect plant growth when discharged into farmland [2]. Treatment of wastewater with high sulfate content produces an acrid smell of sulfur dioxide that can affect air quality and inhibit the elimination of refractory substances [3]. As a result, the growing amount of high-concentration sulfate wastewater must be appropriately managed in terms of harmlessness and resource utilization.

Typically, industrial wastewater with a high salt content or high sulfate concentration can be defined as having total dissolved solids (TDS) greater than or equal to 3.5% (*w*/*w*) [4]. Most high-salt wastewater contains Cl^−^, SO_4_^2−^, Na^+^, Ca^2+^, K^+^, etc. [5]. In the 1970s, developed countries began enforcing legislation prohibiting high-salinity wastewater discharges into water systems, and factories implemented zero-discharge policies [6]. Local governments in China, such as Beijing, Shanghai and Shandong, have introduced standards for measuring TDS in external sewage. Therefore, the harmless and resource-based treatment of high-salt wastewater will become an inevitable trend of water environmental protection. Zero discharge of high-salt wastewater is the subsequent treatment of the wastewater mentioned above through pretreatment, evaporation, crystallization, and other processes so that the recovery rate is higher than 95% [5]. At the same time, crystalline salt is treated in a harmless and resourceful way. NaCl and Na_2_SO_4_ account for 90% of the total inorganic salt in the high-salt wastewater from industries [4]. Thus, the resource utilization of crystalline salt with zero discharge of high-salt wastewater refers to NaCl and Na_2_SO_4_. Specifically, this paper focuses on utilizing high-concentration wastewater sulfate that has caused significant environmental harm.

In recent years, sulfate wastewater treatment technologies have shown signs of a trend of diversification and rapid development, following the promulgation of relevant salt-containing wastewater control standards, ongoing in-depth research around the world, and the emergence of new materials. Once a large area of sulfate contamination has formed, it will be tough to treat it because it is latent and challenging. Therefore, it is necessary to reduce sulfate discharge into the environment. Current sulfate treatment technologies can utilize sulfate wastewater as a resource to a certain extent [7]. Nevertheless, China has not yet developed a standardized recycling system for sulfate wastewater resources, and government supervision of the recycling market is insufficient. The high concentration of sulfate wastewater resources is generally piled-up or discarded randomly, wasting resources and polluting the environment.

The multiple equilibrium problem can be solved well by considering evolutionary game theory’s bounded rationality and learning mechanisms and focusing on decision-making [8]. This article studies the impact of tripartite behaviour changes on sulfate-rich wastewater recovery based on a tripartite evolutionary game model of governments, wastewater recyclers, and wastewater generators. In order to achieve the research goals, we focus on answering the following research questions:(1)What is the return matrix of the three participants in an evolutionary game model for SRW?(2)What are the tripartite evolutionary game model’s equilibrium points and evolutionary stability strategies?(3)What are the conditions for each evolutionary stability strategy?(4)How do changes in the main parameters of the game model affect the behaviour of the three participants?

In order to solve the above problems, our study constructed a tripartite evolutionary game model including the governments, SRW recyclers, and SRW producers, as well as calculated the replication factors dynamic equation for each participant. A Jacobian matrix is then used to obtain the evolutionary stability strategy and conditions. Finally, the model is tested for rationality using numerical simulation of three optimal evolutionary stability strategies of high-concentration sulfate wastewater resource recovery. This paper also discusses changes in parameters affecting the behaviour of three tripartite participants and their behavioural evolution.

In order to solve the above problems, our study used evolutionary game theory to construct a tripartite evolutionary game model including the governments, SRW recyclers, and SRW producers, as well as to calculate the replication factors dynamic equation for each participant. A Jacobian matrix is then used to obtain the evolutionary stability strategy and conditions. Finally, the model is tested for rationality using numerical simulation of three optimal evolutionary stability strategies of high-concentration sulfate wastewater resource recovery. This paper also discusses changes in parameters affecting the behaviour of three tripartite participants and their behavioural evolution.

## 2. Literature Review

This article involves two research streams: sulfate-rich wastewater management and the application of evolutionary game theory (EGT). In industrial sulfate wastewater treatment, there are four main methods: chemical precipitation method, a physical adsorption method, membrane separation method, and biological method [9]. If sulfate wastewater is not managed correctly, it may cause severe environmental and ecological problems. However, if treated as a resource, it can contribute to the industrial ecology and realize the zero-emission mode and the needs of the contemporary development of a circular economy [10]. Many countries and governments have formulated and implemented policies on zero discharge of such industrial high-saline wastewater. For example, in the 1970s, the United States compulsorily stipulated that zero discharge must be implemented due to the impact of industrial wastewater on river water quality [11,12], which is also the world’s earliest zero-discharge policy. Australia’s first zero-discharge industrial wastewater project is also enforced because of policy regulations [13]. Even this year, China’s National Development and Reform Commission and other ten departments jointly issued the “Guiding Opinions on Promoting the Utilization of Sewage Resources,” and zero discharge of industrial wastewater as an important way to achieve the utilization of sewage resources was highlighted [14]. At the same time, other governments are also vigorously encouraging the harmless treatment and resource utilization of industrial wastewater, such as sulfate wastewater. 

In addition to the government authorities’ active promotion of high-concentration sulfate industrial wastewater management, the academic community has also shown great interest in this hot issue. Recent research on sulfate wastewater management has focused on all aspects of this problem. These include sulfate conversions [9,15], technologies for removal of sulfate from wastewater [16,17,18], and reactor model simulation [19,20,21]. Moreover, acid mine drainage (AMD) treatment has always been a hot topic. Many scholars used sulfate-reducing bacteria (SRB) to treat sulfate radicals in AMD, which treat sulfate radicals as resources. For example, Xu et al. [22] use an expanded granular sludge bed reactor (EGSB) and by-product S^2−^ to achieve simultaneous desulfurization and denitrification. Costa et al. [23] use an anaerobic sequencing batch reactor to reduce SO_4_^2−^, and use the produced S^2−^ to remove Fe^2+^, Zn^2+^, and Cu^2+^ in acid mine wastewater, to achieve the recovery and reuse of reaction by-product S^2−^.

However, the studies mentioned above mainly focused on the visible aspects of sulfate wastewater management. Few studies considered the mechanism of sulfate wastewater management from the interaction of behavioural strategies between participants. More importantly, many companies producing sulfate-rich wastewater often distil and crystallize the wastewater, but they cannot dispose of the crystalline product and often discard it. In order to fill this knowledge gap, this article firstly introduces the concept of ‘SRW Recyclers’ and will use evolutionary game theory (EGT), which has proven to be a promising method of analyzing the interaction of multi-agent behaviour strategies. The basic concept of EGT is that the interaction between the individuals in the group is a dynamic process of movement and counter-attack, intertwining an ever-changing gaming environment [24]. Participants are assumed to be entirely rational in the traditional classical game model, and the evolutionary game reverses this hypothesis. When evolutionary play theory is introduced into the phenomenon of biological evolution, the organism is considered to be the participant with limited rationality [24]. In the process of mutual competition, the entities perform the purification of the population, which gives a reasonable explanation for the formation of the habits of specific biological populations. In the 1980s, many economists introduced evolutionary game theory into the economic realm to analyze changes in the social system, industrial developments, and stock markets [25]. At the same time, research on evolutionary game theory has also begun to shift from symmetrical games to asymmetrical games [26]. Foster and Yong [27] introduced continuous random variables to a dynamic system for the first time, and Weibull [28] summed up evolutionary game theory. With the development and improvement of EGT for decades, in the 21st century, many scholars apply evolutionary game theory to the supply chain field [29,30], financial constraint field [31,32], as well electricity market field of research [33].

Recently, EGT expanded its application to waste management and environmental governance. The EGT tool is commonly used to analyze waste management decision-making issues [34]. Waste management with EGT includes research on the recycling of construction waste, electronic waste, and food waste. Some researchers have established a dynamic evolution game model to study how government incentive policies affect the dynamic evolution process of China’s construction waste recycling [35]. Wang et al. [34] proposed a three-party evolutionary game model composed of the governments, recyclers, and consumers to determine the revenue matrix of the e-waste recycling system. In order to reduce food waste and achieve sustainable development, scholars use evolutionary game theory to resume the strategic interaction profit matrix between local governments and supermarkets, revealing the strategic evolutionary choices of both parties [36]. Environmental governance with EGT includes research on environmental policies [37,38,39], environmental supervision [40,41], and pollution prevention [42,43], etc. 

Inspired by previous work, researchers in this study speculate that sulfate-rich wastewater (SRW) resources can be efficiently managed using micro-strategy interaction theory, which considers three participants (i.e., SRW generation department, SRW treatment company, and governments department), and uses evolutionary games in combination with the method. In light of in-depth scenario simulation analysis, some heuristic conclusions are drawn, along with policy recommendations on managing sulfate-rich wastewater. This article makes the following contributions: (1) Most previous studies have focused on two-party games, which addresses the expansion of producer responsibility in three-party games. This study aimed to clarify the primary factors and mechanisms of interaction between participants’ behavioural decisions. (2) The concept of ‘SRW Recyclers’ was first introduced. Such recyclers must experience the sulfate-rich wastewater produced by various industries and conduct resource recovery treatment. (3) This paper focuses on the impact of government intervention and the ecological behaviour of wastewater producers on the behavioural decisions of recyclers.

## 3. Model Building

Developing the sulfate-rich wastewater resource recovery industry is a dynamic process. Under the assumptions of the model, each participant in the system will make different decisions due to changing cost and benefit factors in the process. The relevant stakeholders have limited rationality [34]. In addition, for decades, evolutionary game theory has been proficient and widely used to analyze the decision changes of participants in the system [44]. Therefore, the use of evolutionary game theory can well analyze the behavioural strategies of major players in the sulfate-rich wastewater resource recovery industry. The next step is to describe the model and make assumptions.

### 3.1. Model Description and Assumptions

The governments, sulfate-rich wastewater (SRW) recyclers, and sulfate-rich wastewater (SRW) producers all play a vital role in the resource recovery system for high-concentration sulfate wastewater. The game strategy of the three parties is as follows: (1)***Governments.*** The government has two regulatory strategies, positive supervision (PS) and negative supervision (NS). Under positive supervision, the government actively supervises the SRW recyclers’ performance in wastewater treatment, severely cracks down on the unqualified treatment behaviour of recyclers, and forces the recyclers to reach the qualified treatment level through fines. Under negative supervision, the government does not supervise any wastewater treatment process and results of the recyclers. The government’s reward and punishment measures are necessary external costs in the early stage of environmental protection.(2)***SRW Producers.*** Sulfate-rich wastewater (SRW) producers play a vital role in the high-concentration sulfate recycling industry, and their behaviour decisions significantly impact the development of the recycling industry. Wastewater producers’ behavioural strategies can be classified as ecological (EB) and non-ecological (NB) behaviours. Ecological behaviour means that wastewater producers are environmentally conscious and actively sell wastewater sulfate resources to qualified SRW recyclers. Non-ecological behaviour represents that the wastewater producer has no environmental consciousness and sells wastewater sulfate resources only to unskilled SRW recyclers. Actually, there may be mixed strategies resulting from limited investment opportunities in pro-ecological technologies. Due to the complex situation of mixed strategies, this article is limited in space and will not be discussed.(3)***SRW Recyclers.*** As the executor of the recycling of high sulfate wastewater, sulfate-rich wastewater (SRW) recyclers have two industrial development strategies, namely qualified treatment (QT) and unqualified treatment (UT). Qualified treatment refers to qualified recyclers who upgrade outdated production lines by introducing updated technologies, purchasing updated equipment, and improving infrastructure to meet the requirements of environmental laws. Quality treatment can significantly increase the recycling rate of sulfate resources and reduce environmental damage from the wastewater treatment process. Unqualified treatment means that wastewater recyclers continue to rely on outdated technology, equipment, and infrastructure and that the disposal process is not compliant with environmental regulations. While upgrading high-concentration sulfate resource recycling technology will improve social and environmental performance, it will increase the financial burden on SRW recyclers. Therefore, the government often needs to provide policies to guide SRW recyclers in improving their wastewater treatment technology. It is worth noting that there may be mixed strategies resulting from limited investment opportunities in pro-ecological technologies. Due to the complexity of the hybrid strategy, this article is limited in space and will not discuss.

The relationship between the above entities is shown in Figure 1. Existing research mainly considers that the relationship between participating entities is antagonistic and competitive (for example, wastewater producers and government regulators). However, the relationship between entities in real life can be mutually cooperative and mutually beneficial, and strategic choices can be influenced by each other.

Based on the above analysis of the dynamic interplay between the three parties, we have formulated the following assumptions. Table 1 presents the definition of the parameters implicated in assumptions.

Taking into account the current practice of sulfate-rich wastewater management in the world, the authors made the following six assumptions based on evolutionary game theory (EGT):

**Assumption** **1.**
*Finite rationality. Governments, wastewater recyclers, and wastewater generators are all finitely rational. They can all learn and adapt to dynamic environmental changes and then adjust and optimize their strategies in an evolutionary game. For example, after wastewater generation, it is difficult for wastewater producers to understand the potential value of wastewater. In contrast, wastewater recyclers have difficulty grasping information such as other substances contained in wastewater.*


In contrast, the government has difficulty in obtaining quantitative data on the environmental hazards and environmental benefits of wastewater, so the three parties need to consider whether or not to carry out wastewater resource utilization in the game, and under the limitation of their knowledge and judgment, the decision-making parties are all finite rational. The three parties will continue continuously. With their knowledge and judgment limitations, the decision-makers are limited rational. The three parties will continuously learn the magic formula, compare beneficial strategies, and gradually form a long-term and stable cooperative relationship.

**Assumption** **2.**
*Replication dynamics. Assuming that the game parties learn from each other slowly, the speed of strategy adjustment is simulated by the mechanism of biological evolution “replication dynamics”, i.e., “replication dynamics formula”.*


**Assumption** **3.**
*Resourcefulness strategy. In actual practice, after wastewater generation, wastewater generating enterprises will have two strategic choices: one is to implement green behaviour and carry out wastewater resourceization, providing it as raw materials to recycling organizations and gaining specific revenue; however, additional costs are needed at this time for distillation and crystallization of wastewater; the second is not to carry out green behaviour, just discharging wastewater at will or placing the obtained crystals at will in the plant, causing certain storage costs.*


**Assumption** **4.**
*At the same time, the wastewater recycling sector has two strategic options: one is to implement wastewater resourceization, receiving wastewater and making it into a usable product, but requiring additional costs to develop technical equipment; the other is not to implement wastewater resourceization and to treat wastewater at will.*


**Assumption** **5.**
*At the same time, the government has two strategic options. One is to supervise and provide appropriate rewards and penalties actively. The second is not to actively regulate and condone non-compliant green practices.*


**Assumption** **6.**
*If the probability of the government choosing positive supervision is x, then 1 − x is the probability of choosing negative supervision. If the probability that a wastewater generator chooses ecological behaviour is y, then 1 − y is the probability of choosing non-ecological behaviour. If the probability that the wastewater recycler chooses qualified treatment is z, then 1 − z is the probability that it chooses non-qualified treatment. The government will give a specific financial benefit to wastewater producers and wastewater recyclers who have adopted ecological behaviours. The total amount of the benefit will be Q. Wastewater producers and recyclers will share this benefit in a particular proportion. At the same time, the government will impose a fixed fine of F_R_ on wastewater recyclers who do not treat wastewater under regulations, and the government will use the fine to compensate wastewater producers who have adopted ecological behaviours.*


Based on the above assumptions, under different behavioural strategy choices, the three-party evolutionary game tree of the government, recyclers, and consumers is shown in Figure 2.

### 3.2. The Payoff Matrix and Game Equilibrium Point

According to Figure 2, the payoff matrices for the government, wastewater recyclers, and wastewater producers under different behavioural strategy choices are in Table 2 and Table 3.

#### 3.2.1. Governments

According to Table 2 and Table 3, we can calculate the government’s expected revenue by choosing positive and negative monitoring strategies, denoted by R_G1_ and R_G2_.
(1)$$\begin{array}{cc} R_{G1} & {=\mathrm{yz}(I}_{1}-{\mathrm{GC}}_{1})\\ & +y(1-{z)(I}_{1}-{\mathrm{GC}}_{3})\\ & +z(1-{y)(I}_{1}-{\mathrm{GC}}_{1})\\ & +(1-y)(1-{z)(I}_{1}-{\mathrm{GC}}_{3}) \end{array}$$
(2)$$\begin{array}{cc} R_{G2} & {=\mathrm{yzI}}_{0}\\ & +y(1-{z)I}_{0}\\ & +z(1-{y)I}_{0}\\ & +(1-y)(1-{z)I}_{0} \end{array}$$

According to Equations (1) and (2), it is known that the average expected return of the government is:(3)$$\bar{R_{G}}=xR_{G1}+\left(1-x\right)R_{G2}$$

The government sector strategy replication dynamic equation is:(4)$$\begin{array}{cc} F\left(G\right) & =\frac{\mathrm{dx}}{\mathrm{dt}}\\ & =x\left(1-x\right)\left(R_{G1}-R_{G2}\right)\\ & =x\left(1-x\right)\left[\left(1-m\right)I_{1}-(1-{z)\mathrm{GC}}_{3}-{\mathrm{zGC}}_{1}\right] \end{array}$$

#### 3.2.2. SRW Producers

The dynamic replication equations for SRW producers under the two strategy choices are as follows:(5)$$\begin{array}{cc} R_{P1} & {=\mathrm{xz}(I}_{2}+\mathrm{nI}-C_{1}+\mathrm{rQ})\\ & +x(1-{z)(I}_{2}-C_{1}{+\mathrm{rQ}+F}_{R})\\ & +z(1-{x)(I}_{2}+\mathrm{nI}-C_{1})\\ & +(1-x)(1-{z)(I}_{2}-C_{1}{+F}_{R}) \end{array}$$
(6)$$\begin{array}{cc} R_{P2} & {=\mathrm{xz}(I}_{2}{+E}_{1}-F_{P})\\ & +x(1-{z)I}_{2}\\ & +z(1-{x)(I}_{2}{+E}_{1}-F_{P})\\ & +(1-x)(1-{z)I}_{2} \end{array}$$

The average expected return for SRW producers is:(7)$$\bar{R_{P}}=yR_{P1}+\left(1-y\right)R_{P2}$$

The differential of the replicated dynamic equation for SRW producers is:(8)$$\begin{array}{cc} F\left(P\right) & =\frac{\mathrm{dy}}{\mathrm{dt}}\\ & =y\left(1-y\right)\left(R_{P1}-R_{P2}\right)\\ & =y\left(1-y\right)\left[{\mathrm{xrQ}+z(\mathrm{nI}+F}_{P}-F_{R}-E_{1}{)+F}_{R}-C_{1}\right] \end{array}$$

#### 3.2.3. SRW Recyclers

The dynamic replication equations for SRW recyclers under the two strategy choices are as follows:(9)$$\begin{array}{cc} R_{R1} & =\mathrm{xy}\left[I_{3}+(1-n)I-C_{2}+(1-{r)Q+\mathrm{GC}}_{2}\right]\\ & +x(1-{y)(I}_{3}-C_{2}+(1-{r)Q+F}_{P}{+\mathrm{GC}}_{2})\\ & +y(1-x)\left[I_{3}+(1-n)I-C_{2}\right]\\ & +(1-x)(1-y)\times {(I}_{3}-C_{2}{+F}_{P}) \end{array}$$
(10)$$\begin{array}{cc} R_{R2} & {=\mathrm{xy}(I}_{3}-F_{R}{+E}_{2})\\ & +x(1-{y)I}_{3}\\ & +y(1-x)\times {(I}_{3}-F_{R}{+E}_{2})\\ & +(1-x)(1-{y)I}_{3} \end{array}$$

The average expected return for SRW recyclers is:(11)$$\bar{R_{R}}=zR_{R1}+\left(1-z\right)R_{R2}$$

The differential of the replicated dynamic equation for SRW recyclers is:(12)$$\begin{array}{cc} F\left(R\right) & =\frac{\mathrm{dz}}{\mathrm{dt}}\\ & =z\left(1-z\right)\left(R_{R1}-R_{R2}\right)\\ & =z\left(1-z\right)\left\{\begin{array}{l} x\left[(1-{r)Q+\mathrm{GC}}_{2}\right]\\ +y\left[(1-{n)I+F}_{R}-E_{2}-F_{P}\right]\\ {+I}_{3}{+F}_{P}-C_{2} \end{array}\right\} \end{array}$$

At this point, the differential Equations (4), (8) and (12) constitute a three-dimensional dynamic system of an evolutionary game between the government, SRW collectors, and SRW recyclers, displayed as Equation (13):(13)$$\left\{ \begin{array}{l} F(G)=x\left(1-x\right)\left[\left(1-m\right)I_{1}-(1-{z)\mathrm{GC}}_{3}-{\mathrm{zGC}}_{1}\right]\\ F(P)=y\left(1-y\right)\left[{\mathrm{xrQ}+z(\mathrm{nI}+F}_{P}-F_{R}-E_{1}{)+F}_{R}-C_{1}\right]\\ F(R)=z\left(1-z\right)\left\{x\left[(1-{r)Q+\mathrm{GC}}_{2}\right]+y\left[(1-{n)I+F}_{R}-E_{2}-F_{P}\right]{+I}_{3}{+F}_{P}-C_{2}\right\} \end{array}\right.$$

Let $\left\{ \begin{array}{l} F(G)=0\\ F(P)=0\\ F(R)=0 \end{array}\right.$, then we can obtain eight pure strategy solutions of the three-dimensional dynamical system, which are (0, 0, 0), (0, 1, 0), (0, 0, 1), (0, 1, 1), (1, 0, 0), (1, 0, 1), (1, 1, 0), (1, 1, 1). These eight equilibrium points constitute the boundaries of the solution of the three-way evolutionary game, which is {(x,y,z)|0≤x≤1;0≤y≤1;0≤z≤1}.

For the evolutionary stabilization strategy of the system, the stability of the eight equilibria can be discussed by analyzing the local stability of the Jacobian matrix of a three-dimensional continuous dynamic system.
(14)$$J=\left[\begin{array}{ccc} (1-2x)\left[(1-{m)I}_{1}-(1-{z)\mathrm{GC}}_{3}-{\mathrm{zGC}}_{1}\right] & 0 & x(1-x)\left[{\mathrm{GC}}_{3}-{\mathrm{GC}}_{1}\right]\\ y(1-y)\mathrm{rQ} & (1-2y)\left[{\mathrm{xrQ}+z(\mathrm{nI}+F}_{P}-F_{R}-E_{1})+F_{R}-C_{1}\right] & y(1-y)\left[{\mathrm{nI}+F}_{P}-F_{R}-E_{1}\right]\\ z(1-z)\left[(1-{r)Q+\mathrm{GC}}_{2}\right] & z(1-z)\left[(1-n)I+F_{R}-E_{2}-F_{P}\right] & (1-2z)\left\{x\left[(1-{r)Q+\mathrm{GC}}_{2}\right]+y\left[(1-n)I+F_{R}-E_{2}-F_{P}\right]{+I}_{3}{+F}_{P}-C_{2}\right\} \end{array}\right]$$

As shown in Equation (14), the Jacobi matrix J can be calculated using the three-dimensional differential dynamics equation in Equation (13).

### 3.3. The Solution of Evolutionary Stability Strategy

In this section, stability analysis is performed for eight equilibrium points in a three-dimensional dynamic system, and the eigenvalues of each equilibrium point can be calculated. Then the evolutionary stability strategy of the system is determined. According to Lyapunov’s first method stability theory, when all the eigenvalues of the Jacobi matrix have negative real parts, the equilibrium point is asymptotically stable; when at least one of the eigenvalues of the Jacobi matrix has a positive real part, the equilibrium point is unstable; when the Jacobi matrix has negative real parts except for the eigenvalues with zero genuine parts, the equilibrium point is located in the critical state, and the sign of the eigenvalues cannot determine the stability.

The parameters of the tripartite evolutionary game model of this system are many and complex, and only some of them are selected for analysis due to the limited space of the paper. In order to facilitate the analysis of the sign of the real part of the different equilibrium points and the stability, it is assumed in advance that $\left(1-m\right)I_{1}-{\mathrm{GC}}_{1}<0$, $\left(1-n\right)I+F_{R}-E_{2}-C_{2}>0$,$\;\mathrm{nI}+F_{P}-E_{1}-C_{1}>0$, which represents the net benefit obtained by the government after taking active supervision to promote the recovery and recycling of high-concentration sulfate wastewater is less than the cost of government incentives. In contrast, the net benefit obtained by the wastewater producers and wastewater recyclers is greater than the respective inputs, followed by assumptions about the uncertain conditions. Finally, the equilibrium point of the stabilization point requires all three eigenvalues to be less than zero. Suppose the positive or negative eigenvalues cannot be judged directly. In that case, a discussion is needed on how to judge the stability of each equilibrium point when assuming that this equation is less than 0 or greater than 0. As shown in Table 4, the stability of each equilibrium point can be known.

When $F_{R}-C_{1}>0$ and $F_{P}-C_{2}>0$, or $F_{R}-C_{1}<0$ and $F_{P}-C_{2}<0$, or $F_{R}-C_{1}>0$ and $F_{P}-C_{2}<0$, or $F_{R}-C_{1}<0$ and $F_{P}-C_{2}>0$, does not affect the identified equilibrium point state in Table 4, that is, these condition changes do not affect the determination of the equilibrium point under this study, so in these four cases, E_7_(0, 1, 1) is ESS, and the strategy taken is (negative supervision, ecological behaviours, qualified treatment).

Meanwhile, according to the life cycle theory, this paper divides the life cycle of the sulfate-rich wastewater recycling industry into three stages: early, middle, and mature [34,45]. In the early stage of industry development, the government faces the pressure of environmental degradation. The governments will promulgate some relevant environmental laws and regulations in terms of government administrative effectiveness to strengthen the recovery of sulfate wastewater resources. In addition, due to the difficulty of collection activities, the SRW production sector does not ecologically treat wastewater as well as SRW recyclers will choose substandard treatment strategies. This stage corresponds to ESS (1, 0, 0). It should be noted that the evolutionary stabilization strategy is simplified to ESS. With improved government policy measures and more active monitoring, the industry will enter the middle stage, where sulfate wastewater producers perform ecological treatment of wastewater and recyclers perform qualified recycling of sulfate wastewater resources, a stage corresponding to ESS (1, 1, 1). Therefore, the government will not need to intervene in the market when the industry grows to a scale that forms a relatively complete system for recycling high-concentration sulfate wastewater resources. The industry will enter a mature stage. At this point, the government will slowly withdraw from the market, and the producers and recyclers will constitute the main body of the market. This stage corresponds to ESS (0, 1, 1) in this paper. Since the research objective of this paper is to promote the sustainable development of the sulfate-rich wastewater resource recovery industry, we choose three evolutionary stability strategies corresponding to the above three industrial development stages for detailed analysis. It is worth noting that at this time, to study the three identified stability points, the pre-assumption conditions will be disregarded.

From Table 4, it can be seen that three inequalities need to be satisfied in the early development stage to reach the stability point E_2_(1, 0, 0). The first inequality is ${\mathrm{GC}}_{3}<\left(1-m\right)I_{1}$, which means that the benefits gained by the government in choosing positive supervision are more significant than the input costs, and the government will undoubtedly choose the positive supervision strategy. The second inequality is $\mathrm{rQ}+F_{R}<C_{1}$, which means that the benefits gained by the wastewater producers if they take ecological actions will be smaller than the costs of the action, which leads to the reluctance of the sulfate wastewater producers to undertake ecological behaviours of wastewater. The third inequality is $\left(1-r\right)Q+{\mathrm{GC}}_{2}+I_{3}+F_{P}<C_{2}$, which means that if the wastewater recycler carries out qualified treatment of wastewater, it will make the benefits much smaller than its input costs, so the sulfate wastewater recycler will choose not to carry out qualified treatment of wastewater.

In the middle stage of development, the sulfate wastewater recycling system has started to take shape and still needs to satisfy three inequalities to meet the stabilization point E_8_(1, 1, 1). The first inequality ${\mathrm{GC}}_{1}<\left(1-m\right)I_{1}$ indicates that the government adopts active supervision when it chooses an active supervision strategy. The benefits obtained are greater than its incentive costs. The second inequality is $C_{1\;}<\mathrm{rQ}+\mathrm{nI}+F_{P}-E_{1}$, when the wastewater producer chooses the ecological treatment strategy, the benefits are more significant than the costs. The third inequality is $C_{2}<\left(1-r\right)Q+{\mathrm{GC}}_{2}-\left(1-n\right)I+F_{R}-E_{2}+I_{3}$, when wastewater recyclers choose to qualify for the treatment of sulfate wastewater resources, the benefits outweigh the input costs, and recyclers tend to choose qualified treatment for sulfate wastewater.

At the maturity stage at the end of developing the sulfate wastewater resource recovery industrial system, the industry is free to develop according to the market rules. If the government participates in the excessive intervention, it will lead to an unnecessary financial burden, for the efficient development of the industry requires the government to gradually withdraw from the market and obtain ESS(0, 1, 1). From Table 4, it can be seen that the conditions of three inequalities need to be satisfied to obtain the optimal response dynamic of E_7_(0, 1, 1). The first inequality $\left(1-m\right)I_{1}<{\mathrm{GC}}_{1}$ shows that the benefits of positive government supervision are already less than the incentive cost of its investment, so the government tends to withdraw from the market and adopt a negative supervision strategy. From the second inequality $C_{1\;}<\mathrm{nI}+F_{P}-E_{1}$, it can be seen that sulfate-rich wastewater producers who adopt ecological treatment will achieve higher benefits than their input costs. From the third inequality $C_{2}<\left(1-n\right)I+F_{R}-E_{2}+I_{3}$, it can be seen that sulfate-rich wastewater recyclers will also achieve benefits higher than their input costs if they adopt qualified treatment of wastewater. Hence, SRW recyclers tend to choose qualified treatment of sulfate wastewater and make the whole industry enter the circular economy system.

## 4. Numerical Simulation

The recovery and recycling industry of high-concentration sulfate wastewater resources has not been systematized and is only in the initial stage. Even some governments are not yet aware of recovering and reusing this resource. In addition, the government with executive power needs to be a necessary force in developing this industry. It is necessary to explore the influence of government incentives and penalties on the willingness of the other two participants to participate in the recovery and recycling of SRW resources. So we conducted the corresponding numerical simulation in MATLAB software, focusing on the dynamic evolution of the willingness of the government, SRW producers, and SRW recyclers to participate in various states. At the same time, a sensitivity analysis of the parameters of the government incentives and penalties is conducted during the numerical simulation to verify the correctness of the theoretical analysis, and a graphical representation is used to compensate for the lack of intuitiveness of the theoretical analysis. This part provides meaningful theoretical guidance for the smooth start and future development of SRW resources recovery and recycling.

### 4.1. Evolution of Three Scenarios for the Early, Middle, and Mature Development Stages of the SRW Resources Recycling Industry

#### 4.1.1. The Early Development Stage E_2_(1, 0, 0)

The initial willingness of the government, SRW producer, and SRW recycler are taken to be *x* = 0.4, *y* = 0.2, and *z* = 0.3, respectively, and the relevant parameters are set as follows: m = 0.5; I_1_ = 8; GC_3_ = 0.6; GC_1_ = 0.6; r = 0.5; Q = 2; n = 0.5; I = 4; F_P_ = 1; F_R_ = 1.5; E_1_ = 2; GC_2_ = 3; E_2_ = 1; I_3_ = 2; C_2_ = 10, C_1_ = 3, and the parameters satisfy the stability determination condition of (1, 0, 0), which also means ${\mathrm{GC}}_{3}-\left(1-m\right)I_{1}<0$, $\;\mathrm{rQ}+F_{R}-C_{1}<0$, $\;\left(1-r\right)Q+{\mathrm{GC}}_{2}+I_{3}+F_{P}-C_{2}<0$, and the evolution results are shown in Figure 3.

As shown in Figure 3, the government eventually becomes an evolutionary stabilization strategy with positive supervision over time in the early development stage. SRW producers and SRW recyclers eventually evolve to non-ecological behaviours and unqualified treatment, respectively, and the rate of evolution is faster for SRW recyclers than SRW producers.

#### 4.1.2. The Middle Development Stage E_8_(1, 1, 1)

The initial willingness of the government, SRW producer, and SRW recycler are taken as *x* = 0.6, *y* = 0.5, and *z* = 0.6, respectively, and the relevant parameters are set as follows: m = 0.5; I_1_ = 8; GC_3_ = 0.6; GC_1_ = 0.3; r = 0.5; Q = 2; n = 0.5; I = 4; F_P_ = 1; F_R_ = 2.5; E_1_ = 1.5; GC_2_ = 0.9; E_2_ = 1; I_3_ = 2; C_2_ = 3; C_1_ = 1, the parameters satisfy the stability determination condition of (1, 1, 1), which represents that ${\mathrm{GC}}_{1}-\left(1-m\right)I_{1}<0$, $-\mathrm{rQ}-\mathrm{nI}-F_{P}+E_{1}+C_{1}<0$, $-\left(1-r\right)Q-{\mathrm{GC}}_{2}+\left(1-n\right)I-F_{R}+E_{2}-I_{3}+C_{2}<0$, and the evolutionary results are shown in Figure 4. According to Figure 4, it can be seen that the strategies of the government, SRW producers, and SRW recyclers converge to positive supervision, ecological behaviours, and qualified treatment, respectively, over time in the middle development stage, but the convergence rate of SRW producers is the slowest. Over time, the government has reached an equilibrium and opted for an active surveillance strategy very quickly. SRW recyclers reached equilibrium relatively quickly and chose ecological behaviour strategies, while SRW producers took the longest to reach equilibrium and finally chose qualified treatment strategies for sulfate-rich wastewater treatment.

#### 4.1.3. The Mature Development Stage E_7_(0, 1, 1)

The initial willingness of the government, SRW producer, and SRW recycler are taken as *x* = 0.7, *y* = 0.7, and *z* = 0.8, respectively, and the relevant parameters are set as follows: m = 0.5, I_1_ = 2, GC_3_ = 0.6, GC_1_ = 2.4, r = 0.5, Q = 2, n = 0.5, I = 4, F_P_ = 10, F_R_ = 2.5, E_1_ = 1.5, GC_2_ = 0.9, E_2_ = 1, I_3_ = 2, C_2_ = 3, C_1_ = 8 and the parameters satisfy the stability determination condition of (0, 1, 1), implying that $\left(1-m\right)I_{1}-{\mathrm{GC}}_{1}<0$, $-\mathrm{nI}-F_{P}+E_{1}+C_{1}<0$, $-\left(1-n\right)I-F_{R}+E_{2}-I_{3}+C_{2}<0$, the evolution results are shown in Figure 5. Over time, the behaviour of SRW producers and SRW recyclers evolved to ecological behaviour and qualified treatment, respectively, while the government’s behaviour evolved to negative supervision. The reason is that over time, in this situation, the government finds that adopting active regulation is not its preferred strategy and shifts to an adverse regulatory strategy in order to maximize its interests. In a nutshell, the probability of the government choosing a positive supervision strategy has been decreasing at this stage, and finally opted for negative supervision. SRW recyclers choose the ecological behaviour strategy and reach equilibrium relatively quickly, while SRW producers take a longer time to reach the equilibrium stage and choose a qualified treatment strategy.

### 4.2. The Effect of Parameter Changes of Government Rewards and Punishments on the Initial Evolutionary Results of Tripartite Subjects

In order to further reveal the effects of changes in model parameters on the evolutionary results, sensitivity analyses were conducted on some critical parameters, including government subsidies GC_2_ for participation in wastewater recycling, government costs GC_3_ without any incentives, financial benefits Q provided by the government, and fines F_P_ imposed by the government on wastewater producers who adopt non-ecological behaviours.

The GC_2_ parameters are set to 3, 6, and 9, respectively, and the evolution results are shown in Figure 6. From Figure 6, it can be seen that the more considerable the number of government subsidies for government participation in wastewater recycling, the more SRW recyclers tend to adopt qualified treatment strategies without changing the evolution rate and evolution trend of government, SRW producers, which is because with the gradual increase of inequality $\left(1-r\right)Q+{\mathrm{GC}}_{2}+I_{3}+F_{P}-C_{2}<0$ is destroyed. At the same time, there is no effect on inequality ${\mathrm{GC}}_{3}-\left(1-m\right)I_{1}<0$, $\mathrm{rQ}+F_{R}-C_{1}<0$, so government and SRW producers still evolve and stabilize to (1, 0), consistent with the similar findings of Su et al. [45].

The evolutionary results are set at GC_3_ to 0.6, 1.2, and 1.8, respectively, shown in Figure 7. It can be seen from Figure 7 that the rate of government evolving into a positive supervision strategy slows down as the government cost without any incentive increases, which indicates that with the increase of supervision cost, the government’s willingness to choose negative supervision increases, and the rate of SRW producers and SRW recyclers evolving into non-ecological behaviour and unqualified treatment gradually accelerates. The high supervision cost may reduce the willingness of government regulators to regulate actively. The high cost of supervision may reduce the willingness of government regulators to regulate actively, thus discouraging SRW producers and SRW recyclers from adopting eco-behaviour and qualified treatment.

Setting Q to 2, 4, and 8, respectively, the evolutionary results are shown in Figure 8. From Figure 8, it can be seen that the government department eventually evolved into a positive supervision strategy in all three cases, which indicates that the change in the financial benefits provided by the government has little effect on the government’s willingness to regulate. SRW producers evolve to adopt non-ecological behaviour when the financial benefits provided by the government are 2. In contrast, SRW producers evolve to adopt ecological behaviour when the financial benefits provided by the government are 4 and 8, which indicates that incentive measures have a positive effect on the adoption of ecological behaviour by SRW producers. On the other hand, comparing Q = 4 and Q = 8, we can also observe that the incentives also positively influence the eligible disposal behaviour of SRW recyclers. In the case of Q = 8, the government provides more financial support, and the unilateral implementation of SRW recycling is less risky. Hence, producers and recyclers are willing to adopt ecological behaviour, and qualified disposal behaviour.

The F_P_ were set to 1, 2, and 8, respectively, and the evolutionary results are shown in Figure 9. From Figure 9, it can be seen that the government tends to eventually adopt active supervision measures in all three cases, which indicates that the government’s attitude toward SRW producers who do not adopt ecological treatment of sulfate wastewater is consistently disapproving. When the government fines for SRW producers are at 1 or 2, both wastewater producers and wastewater recyclers do not tend to adopt ecologically beneficial behaviours, and the government’s punitive measures do not have the intended effect. However, once the government fines were raised to 8, the wastewater producers became inclined to adopt ecological behaviours.

## 5. Conclusions

In this paper, in order to explore how to solve the problem of recycling and treatment of high-concentration sulfate wastewater, we guide the construction of an evolutionary game model of the government, SRW recyclers, and SRW producers, analyze the long-term learning behaviours, and strategy adjustment mechanisms of the three participants, identify three stability states that can represent the SRW recycling industry and theorize them and finally draw some meaningful conclusions through visual numerical simulations.

The results suggest that the government should play a leading role in developing the SRW resource recovery industry. In the early stage of industry development, the government should increase subsidies to recyclers and appropriately control supervision costs. The government should adjust the regulatory mechanism of rewards and penalties to promote the development of the sulfate wastewater recycling industry to the middle stage. At the mid-stage, the government should control the subsidy level, and appropriate subsidies can help SRW producers adopt ecological behaviour strategies for sulfate wastewater. SRW producers protect the environment in the mature stage, and recyclers actively collect and recover compliant sulfate wastewater resources. Governments should gradually deregulate and eventually withdraw from the market. Qualified recyclers and environmentally friendly wastewater producers can benefit from a mature SRW resources recovery industry. If government subsidies are too low, SRW recyclers will be reluctant to choose ecological behaviour strategies due to the high cost they need to bear. If government subsidies are too high, this will create a financial burden for the government, and it will be detrimental for the government to take active regulatory measures. In fact, it can be recommended that the government try its best to reduce the regulatory cost of high-concentration sulfate wastewater recovery so that subsidies can be increased within reasonable limits. In addition, the government’s fines for unreasonable behaviours will effectively prevent the non-ecological behaviour of SRW producers and SRW recyclers, and promote the formation of the sulfate wastewater recycling industry chain.

In this paper, dismantling qualified SRW recyclers will incur irreversible investment costs due to introducing new equipment and upgrading production lines. The model in this paper ignores the impact of irreversible sunk investment costs on strategy choice. Therefore, we will incorporate the actual costs into the sunken irreversible investment costs in future research and develop an evolutionary game model that includes the actual costs. We hope to effectively reduce the cost risk of SRW recyclers and better promote the sustainable development of the resource-based recycling industry for high-concentration sulfate wastewater. Since some assumptions are required before the model is established, the research has reached the correct conclusion as a whole. If we delve into the details of mixed strategies or investment economic scale, it will lead to new thinking. Such considerations will be discussed in subsequent studies.

Another limitation of this paper is that we only model the coordination of stakeholders’ interests in the SRW resource recovery process by setting stochastic parameters, mainly due to the lack of an accurate set of data. In addition, we only consider government incentives and penalties in SRW recycling. In future studies, we will consider incorporating as many influencing factors as possible into the framework and using accurate data to examine the coordination of stakeholders’ interests in SRW resource recovery.

## Figures and Tables

**Figure 1 ijerph-19-08770-f001:**
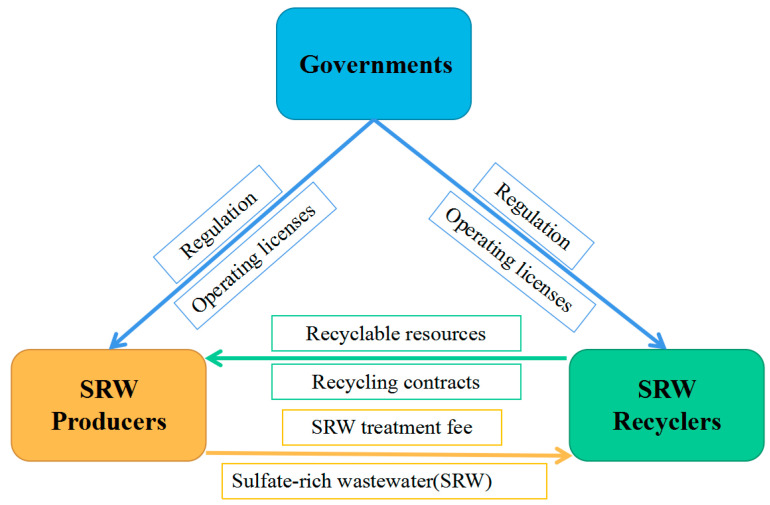
Relationship among the participants in sulfate-rich wastewater (SRW) management.

**Figure 2 ijerph-19-08770-f002:**
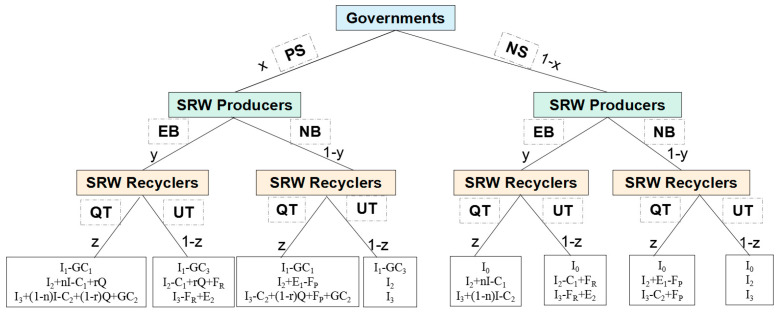
The three-party evolutionary game tree.

**Figure 3 ijerph-19-08770-f003:**
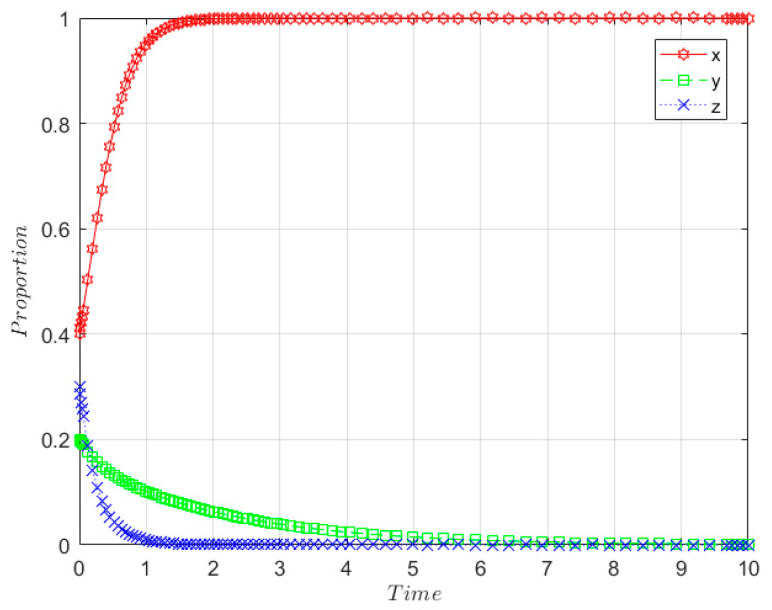
Evolution path of stability point E_2_(1, 0, 0) in the early development stage.

**Figure 4 ijerph-19-08770-f004:**
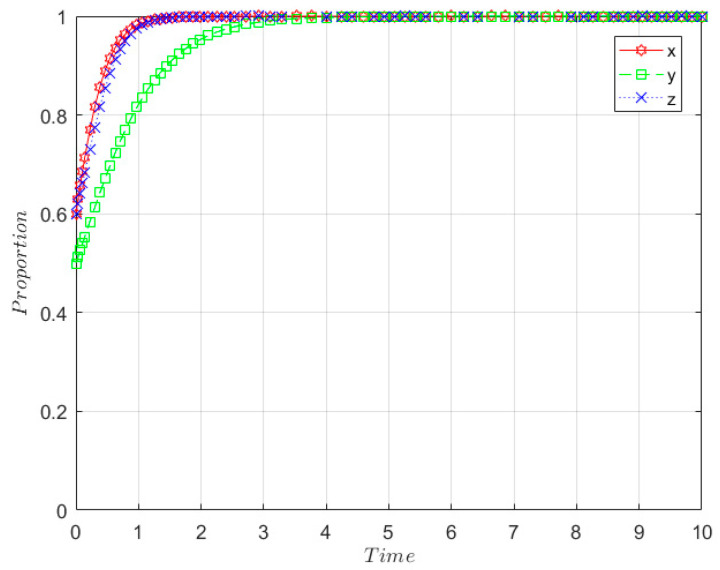
Evolution path of stability point E_8_(1, 1, 1) in the middle development stage.

**Figure 5 ijerph-19-08770-f005:**
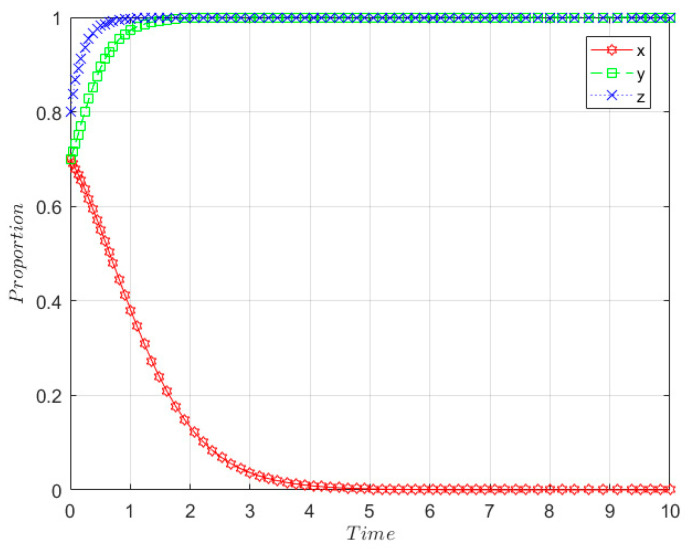
Evolution path of stability point E_7_(0, 1, 1) in the mature development stage.

**Figure 6 ijerph-19-08770-f006:**
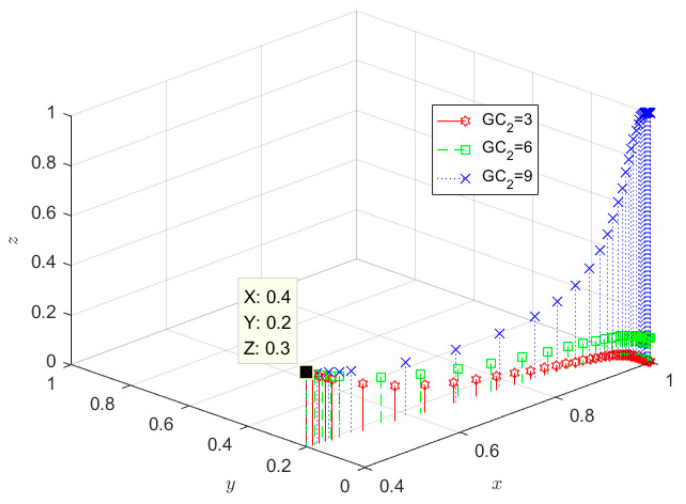
Impact of government subsidies for participating in the recycling of wastewater.

**Figure 7 ijerph-19-08770-f007:**
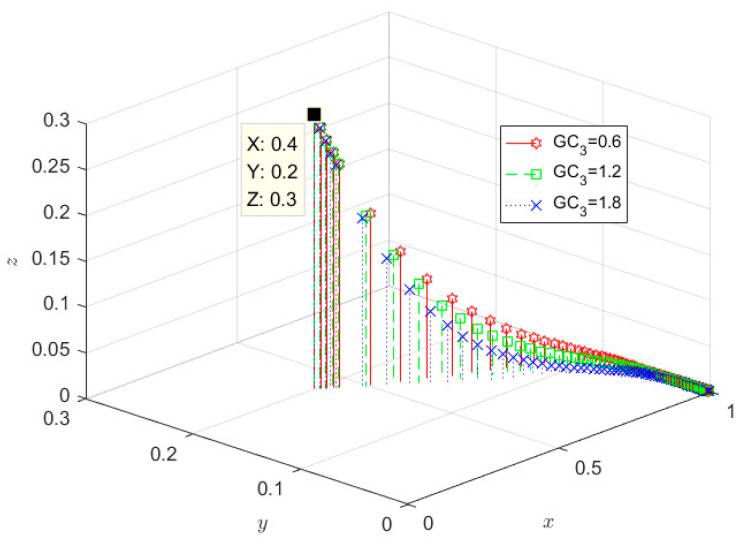
Impact of the cost of the government without any incentives.

**Figure 8 ijerph-19-08770-f008:**
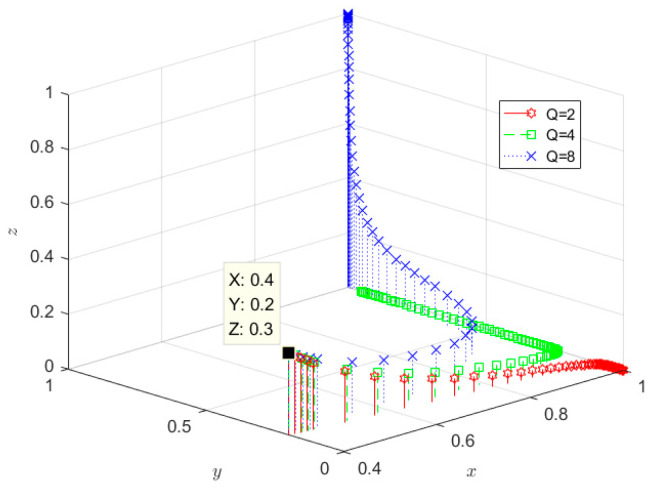
Impact of financial benefits is given by the government.

**Figure 9 ijerph-19-08770-f009:**
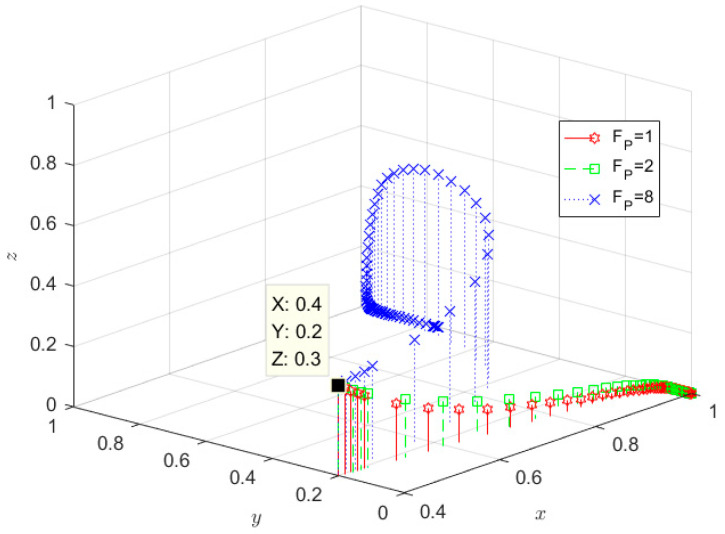
Impact of government fines for wastewater producers who take non-ecological behaviours.

**Table 1 ijerph-19-08770-t001:** Definition of parameters.

Game Subject	Parameter	Definition	Range
Governments	x	Governments choose positive supervision (PS)	[0,1]
	1 − x	Governments choose negative supervision (NS)	[0,1]
SRW Producers	y	SRW producers choose ecological behaviours (EB)	[0,1]
	1 − y	SRW producers choose non-ecological behaviours (NB)	[0,1]
SRW Producers	z	SRW recyclers choose qualified treatment (QT)	[0,1]
	1 − z	SRW recyclers choose unqualified treatment (UT)	[0,1]
Governments	GC_1_	The cost of government incentives	(0,+∞)
	GC_2_	Government subsidies for participating in the recycling of wastewater	(0,+∞)
	GC_3_	The cost of the government without any incentives	(0,+∞)
	Q	Financial benefits are given by the government	(0,+∞)
	I_0_	Benefits when the government adopts passive supervision	(0,+∞)
	I_1_	Benefits when the government adopts active supervision	(0,+∞)
	m	The ratio of government revenue in the two cases, where m = I_0_/I_1_	[0,1]
SRW Producers and Recyclers	C	The total cost of wastewater producers and wastewater recyclers, where C = C_1_ + C_2_	(0,+∞)
	I	Total additional benefits for producers and recyclers involved in wastewater recycling	(0,+∞)
SRW Producers	C_1_	The cost of wastewater producers taking ecological actions on wastewater	(0,+∞)
	r	The cost of wastewater producers as a percentage of the total cost, where r = C_1_/C	[0,1]
	I_2_	The benefits of wastewater producers not participating in wastewater ecological behaviour	(0,+∞)
	n	Percentage of wastewater producer’s additional revenue as a percentage of total additional revenue	[0,1]
	E1	When the recycler participates and the producer does not participate, the producer gains benefits	(0,+∞)
	FP	Government fines for wastewater producers who take non-ecological behaviours	(0,+∞)
SRW Producers	C_2_	The cost of wastewater recyclers for quality treatment of wastewater	(0,+∞)
	I_3_	Wastewater recyclers do not participate in the benefits of qualified treatment	(0,+∞)
	E2	When the producer participates and the recycler does not participate, the recycler gains benefits	(0,+∞)
	FR	Government fines for wastewater recyclers who take substandard treatment	(0,+∞)

Note: SRW (sulfate-rich wastewater).

**Table 2 ijerph-19-08770-t002:** Payoff matrix with governments that choose positive supervision.

		SRW Recyclers	
		Qualified Treatment	Non-Qualified Treatment
SRW Producers	Ecological behaviour	I_1_ − GC_1_	I_1_ − GC_3_
		I_2_ + nI − C_1_ + rQ	I_2_ − C_1_ + rQ + F_R_
		I_3_ + (1 − n)I − C_2_ + (1 − r)Q + GC_2_	I_3_ − F_R_ + E_2_
	Non-ecological behaviour	I_1_ − GC_1_	I_1_ − GC_3_
		I_2_ + E_1_ − F_P_	I_2_
		I_3_ − C_2_ + (1 − r)Q + F_P_ + GC_2_	I_3_

**Table 3 ijerph-19-08770-t003:** Payoff matrix with governments that choose negative supervision.

		SRW Recyclers	
		Qualified Treatment	Non-Qualified Treatment
SRW Producers	Ecological behaviour	I_0_	I_0_
		I_2_ + nI − C_1_	I_2_ − C_1_ + F_R_
		I_3_ + (1 − n)I − C_2_	I_3_ − F_R_ + E_2_
	Non-ecological behaviour	I_0_	I_0_
		I_2_ + E_1_ − F_P_	I_2_
		I_3_ − C_2_ + F_P_	I_3_

**Table 4 ijerph-19-08770-t004:** Equilibrium point of the system and eigenvalues.

Equilibrium Point	Eigenvalues		Real Part Notation	Stability Conclusion
E_1_(0, 0, 0)	λ_1_	$\left(1-m\right)I_{1}-{\mathrm{GC}}_{3}$	−	Unstable
	λ_2_	$F_{R}-C_{1}$	×	
	λ_3_	$I_{3}+F_{P}-C_{2}$	+	
E_2_(1, 0, 0)	λ_1_	${\mathrm{GC}}_{3}-\left(1-m\right)I_{1}$	+	Unstable
	λ_2_	$\mathrm{rQ}+F_{R}-C_{1}$	×	
	λ_3_	$\left(1-r\right)Q+{\mathrm{GC}}_{2}+I_{3}+F_{P}-C_{2}$	+	
E_3_(0, 1, 0)	λ_1_	$\left(1-m\right)I_{1}-{\mathrm{GC}}_{3}$	−	Unstable
	λ_2_	$C_{1}-F_{R}$	×	
	λ_3_	$\left(1-n\right)I+F_{R}-E_{2}+I_{3}-C_{2}$	+	
E_4_(0, 0, 1)	λ_1_	$\left(1-m\right)I_{1}-{\mathrm{GC}}_{1}$	−	Unstable
	λ_2_	$\mathrm{nI}+F_{P}-E_{1}-C_{1}$	+	
	λ_3_	$C_{2}-I_{3}-F_{P}$	×	
E_5_(1, 1, 0)	λ_1_	${\mathrm{GC}}_{3}-\left(1-m\right)I_{1}$	+	Unstable
	λ_2_	$C_{1}-\mathrm{rQ}-F_{R}$	×	
	λ_3_	$\left(1-r\right)Q+{\mathrm{GC}}_{2}+\left(1-n\right)I+F_{R}-E_{2}+I_{3}-C_{2}$	+	
E_6_(1, 0, 1)	λ_1_	${\mathrm{GC}}_{1}-\left(1-m\right)I_{1}$	+	Unstable
	λ_2_	$\mathrm{rQ}+\mathrm{nI}+F_{P}-E_{1}-C_{1}$	+	
	λ_3_	$C_{2}-\left(1-r\right)Q-{\mathrm{GC}}_{2}-I_{3}-F_{P}$	×	
E_7_(0, 1, 1)	λ_1_	$\left(1-m\right)I_{1}-{\mathrm{GC}}_{1}$	−	ESS
	λ_2_	$-\mathrm{nI}-F_{P}+E_{1}+C_{1}$	−	
	λ_3_	$-\left(1-n\right)I-F_{R}+E_{2}-I_{3}+C_{2}$	−	
E_8_(1, 1, 1)	λ_1_	${\mathrm{GC}}_{1}-\left(1-m\right)I_{1}$	+	Unstable
	λ_2_	$-\mathrm{rQ}-\mathrm{nI}-F_{P}+E_{1}+C_{1}$	−	
	λ_3_	$-\left(1-r\right)Q-{\mathrm{GC}}_{2}+\left(1-n\right)I-F_{R}+E_{2}-I_{3}+C_{2}$	−	

Note: × means the real part notation is uncertain.

## Data Availability

Not applicable.

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
