# Peer review of "Multi-Agent Evolutionary Game in the Recycling Utilization of Sulfate-Rich Wastewater"

_ijerph, 2022, doi:10.3390/ijerph19148770_

Round 1

Reviewer 1 Report

Introduction should be improved. Authors in last paragraph added information what will be written in the next parts. It is not necessary. In the end of introduction will be better if authors provided aim of research.

Lack of numerical form of results. What differences will be if authors used other wastes to analysis, for example nitrogen? Presentation of sulfate rich wastewater utilization is poor. I suggest improving this part. Presented on figure 1 Relationship among the participants in sulfate-rich wastewater (SRW) management. In my opinion is too simply. What about if users of recyclable  resources are other participants not SRW. Presented in manuscript proposal of use the mechanism of evolutionary game theory (EGT) to conduct theoretical modelling and simulation analysis on the interaction of the behavior of the above three participants is interesting. But in fact external cost of environmental protection. It is basic situation. In conclusion authors have wrote: The results suggest that the government should play a leading role in developing the SRW resource recovery industry. But what authors expected? Who others can do this. Please improve of conclusion be adding some important values taken from results part.

Author Response

Response to Reviewer #1

[Reviewer#1-General comments]

Introduction should be improved. Authors in last paragraph added information what will be written in the next parts. It is not necessary. In the end of introduction will be better if authors provided aim of research.

Response: Thank you very much. To make it easier for readers, we have incorporated part of the literature review in the introduction into the next section, and we have trimmed the purpose of the study for clarity.[Pg30-100]

[Reviewer#1-Specific comment 1] Lack of numerical form of results. What differences will be if authors used other wastes to analysis, for example nitrogen? Presentation of sulfate rich wastewater utilization is poor. I suggest improving this part.

Response: Thanks for your kind reminders. The article focuses on management strategies, and there are few numerical results. The results are mainly quantified by formulas and graphics. We will consider the reviewer's interest in nitrogen in our follow-up research. We mention this necessary information in the article.‘Some past studies mainly focused on the visible aspects of sulfate wastewater management. Few studies considered the mechanism of sulfate wastewater management from the interaction of behavioural strategies between participants.‘[Pg137-140]

[Reviewer#1-Specific comment 2] Presented on figure 1 Relationship among the participants in sulfate-rich wastewater (SRW) management. In my opinion is too simply. What about if users of recyclable resources are other participants not SRW.

Response: Thanks for your kind reminders. If the real management of SRW is considered, it will indeed involve many social roles. In addition to the government, SRW producers and SRW recyclers, there are many roles that will affect the management of SRW. As far as this article is concerned, our purpose is to discuss SRW resource recycling issues, and the concept of SRW recyclers will inevitably be introduced, and the most important role in connecting with SRW recyclers is the government and SRW producers. Of course, the reviewer have brought up good angles, and follow-up research will delve deeper into the role of other social roles in SRW management. [Pg140-145]

[Reviewer #1-Specific comment 3] Presented in manuscript proposal of use the mechanism of evolutionary game theory (EGT) to conduct theoretical modelling and simulation analysis on the interaction of the behavior of the above three participants is interesting. But in fact external cost of environmental protection. It is basic situation.

Response: Thanks for your kind reminders. We agree with the reviewer's idea and include it in the article as a necessary clarification.[Pg205-209]

[Reviewer#1-Specific comment 4] In conclusion authors have wrote: The results suggest that the government should play a leading role in developing the SRW resource recovery industry. But what authors expected? Who others can do this. Please improve of conclusion be adding some important values taken from results part.

Response: Thanks for your kind reminders. We have improved the conclusion part according to the reviewer’s suggestion. [Pg591]

Reviewer 2 Report

General comments:

The submitted work applies multi-agent evolutionary game theory (EGT) to study the interactive effects among the industries generating sulfate-rich wastewater (SRW), their recycling process, and the government initiatives in developing the sulfate recovery industry thus, accentuating resource circular economy. The study, in general, is quite interesting. However, several issues need to be addressed before its further processing.

Specific comments:

1.  The language of the manuscript needs significant improvement. Coherence is missing in many cases, along with inappropriate word choice.

2. Consider rewriting the abstract. In the abstract section, the authors emphasized the problem statement while the findings and implications of the study were not highlighted enough.  

1      3. Better to merge the introduction and literature review section. Having two separate sections makes the article lengthy, and the readers' focus can be easily lost. The authors should present the background of the study, relevant literature, and problem statement in a robust way. Also, write the study objectives more precisely.

     4. Include some information on forms and applications of recovered sulfate from waste streams.

     5. L40: Write the ionic forms correctly. Check for such issues throughout the manuscript.

     6. The section ‘Model building, provides sufficient information and is well-structured.

     7. The results of the numerical simulation should be discussed comprehensively for a better understanding of the readers.

     8. Check the number of the conclusion section.

4

Author Response

Response to Reviewer #2

[Reviewer #2-General comments]

The submitted work applies multi-agent evolutionary game theory (EGT) to study the interactive effects among the industries generating sulfate-rich wastewater (SRW), their recycling process, and the government initiatives in developing the sulfate recovery industry thus, accentuating resource circular economy. The study, in general, is quite interesting. However, several issues need to be addressed before its further processing.

Response: Thank you very much. We have revised the mentioned issues in the article.[All content]

[Reviewer #2-Specific comment 1] The language of the manuscript needs significant improvement. Coherence is missing in many cases, along with inappropriate word choice.

Response: Thanks for your kind reminders. We have made improvements to the manuscript language and made changes to inappropriate word choices.[All content]

[Reviewer #2-Specific comment 2] Consider rewriting the abstract. In the abstract section, the authors emphasized the problem statement while the findings and implications of the study were not highlighted enough.

Response: Thanks for your kind reminders. We have rewritten the Abstract with further emphasis on the findings and implications of the study.

‘Abstract: Current industrial development has led to an increase in sulfate-rich industrial sewage, threatening industrial ecology and the environment. Incorrectly treating high-concentration sulfate wastewater can cause serious environmental problems and even harm human health. Water with high sulfate levels can be treated as a resource and treated harmlessly to meet the needs of the circular economy. Today, governments worldwide are working hard to encourage the safe disposal and reuse of industrial salt-rich wastewater by recycling sulfate-rich wastewater (SRW) resources. However, the conflict of interests between the SRW production department, the SRW recycling department, and the government often make it challenging to effectively manage sulfate-rich wastewater resources. This study aims to use the mechanism of evolutionary game theory (EGT) to conduct theoretical modelling and simulation analysis on the interaction of the behaviour of the above three participants. This paper focuses on the impact of government intervention and the ecological behaviour of wastewater producers on the behavioural decisions of recyclers.The results suggest that the government should play a leading role in developing the SRW resource recovery industry. SRW producers protect the environment in the mature stage, and recyclers actively collect and recover compliant sulfate wastewater resources. Governments should gradually deregulate and eventually withdraw from the market. Qualified recyclers and environmentally friendly wastewater producers can benefit from a mature SRW resources recovery industry.’ [Abstract]

[Reviewer #2-Specific comment 3] Better to merge the introduction and literature review section. Having two separate sections makes the article lengthy, and the readers' focus can be easily lost. The authors should present the background of the study, relevant literature, and problem statement in a robust way. Also, write the study objectives more precisely.

Response: Thanks for your kind reminders. To make it easier for readers, we have incorporated part of the literature review in the introduction into the next section, and we have trimmed the purpose of the study for clarity.[Pg30-100]

[Reviewer #2-Specific comment 4] Include some information on forms and applications of recovered sulfate from waste streams.

Response: Thanks for your kind reminders. We have highlighted and added some infromation and applications of recovered sulfate from waste streams. [Pg 103-106, Pg126-136 ]

[Reviewer #2-Specific comment 5] L40: Write the ionic forms correctly. Check for such issues throughout the manuscript.

Response: Thanks for your kind reminders. We have checked for ionic forms throughout the manuscript. [Pg44-45,57,648]

[Reviewer #2-Specific comment 6] The section ‘Model building, provides sufficient information and is well-structured.

Response: Thank you very much.

[Reviewer #2-Specific comment 7] The results of the numerical simulation should be discussed comprehensively for a better understanding of the readers.

Response: Thanks for your kind reminders. We have discussed the numerical simulation results further. [Pg465]

[Reviewer #2-Specific comment 8] Check the number of the conclusion section.

Response: Thanks for your kind reminders. We have checked the number of the conclusion section.[Pg591]

Reviewer 3 Report

The paper discusses the modeling of the behavior of three parties involved in the production and recycling of sulphates contained in industrial wastewater. The authors analyze the operation of the government side, i.e. legislation in regulation, and encourage producers and recyclers to act pro-ecologically. The analysis shows that only financial incentives and penalties can induce pro-ecological behavior, and from a certain level of invested investment of both parties, no restrictive measures will be needed to maintain pro-ecological activities of both parties. In my opinion, this is a wrong conclusion because both parties will always try to reduce the costs of their activity.

The work is written in clear language, although there are fragments with grammatical and editorial errors.
Line 40 SO42- no upper or lower index
Line 43-44 Chinese local standards ... have local standards for ...
Lines 27-97 Introduction already contain Literature review
Lines 192-194 is only a guess, not an established fact
Line 200 and in several places use the plural "governments" instead of the singular.
Line 210 The authors suggest that SRW producers may use pro-ecological or ecological strategies and do not mention that there may be mixed strategies resulting from limited investment opportunities in pro-ecological technologies. The same is true for SRW Recyclers (line 217)
Line 244 "Author" used singular, while the manuscript has two authors.
Lines 283-287 "1-x", y, z and other symbols should be in italics "1-x", y, z
Line 291 FR should be subscript  FR
Line 363-364 "parameters" ... "are many" - something is wrong with the grammar
Line 390 "production SRW sector" should be "SRW production sector"

Author Response

Response to Reviewer #3

[Reviewer #3-General comments]

The paper discusses the modeling of the behavior of three parties involved in the production and recycling of sulphates contained in industrial wastewater. The authors analyze the operation of the government side, i.e. legislation in regulation, and encourage producers and recyclers to act pro-ecologically. The analysis shows that only financial incentives and penalties can induce pro-ecological behavior, and from a certain level of invested investment of both parties, no restrictive measures will be needed to maintain pro-ecological activities of both parties. In my opinion, this is a wrong conclusion because both parties will always try to reduce the costs of their activity.

Response: Thank you very much. In response to the reviewer's thoughts, we have added some information to the conclusion:’Since some assumptions are required before the model is established, the research has reached the correct conclusion as a whole. If we delve into the details of mixed strategies or investment economic scale, it will lead to new thinking. Such considerations will be discussed in subsequent studies’.[Pg608-612]

[Reviewer #3-Specific comment 1] Line 40 SO42- no upper or lower index

Response: Thanks for your kind reminders. We have improved this issue, changed SO42- to  SO42-.[Pg44]

[Reviewer #3-Specific comment 2] Line 43-44 Chinese local standards ... have local standards for ...

Response: Thanks for your kind reminders. We have changed this sentence to ‘Local governments in China, such as Beijing, Shanghai and Shandong, have introduced standards for measuring TDS in external sewage.’ We hope that this rewriting will make the meaning of the sentences clearer.[Pg47]

[Reviewer #3-Specific comment 3] Lines 27-97 Introduction already contain Literature review

Response: Thanks for your kind reminders. To make it easier for readers, we have incorporated part of the literature review in the introduction into the next section.[Pg30-100]

[Reviewer #3-Specific comment 4] Lines 192-194 is only a guess, not an established fact

Response: Thanks for your kind reminders. We have added model assumptions to this sentence. [Pg196-197]

[Reviewer #3-Specific comment 5] Line 200 and in several places use the plural "governments" instead of the singular.

Response: Thanks for your kind reminders. The singular and plural forms of "government" have been unified.[Pg205-209]

[Reviewer #3-Specific comment 6] Line 210 The authors suggest that SRW producers may use pro-ecological or ecological strategies and do not mention that there may be mixed strategies resulting from limited investment opportunities in pro-ecological technologies. The same is true for SRW Recyclers (line 217)

Response: Thanks for your kind reminders. We have provided additional clarification on this situation. [Pg218-246]

[Reviewer #3-Specific comment 7] Line 244 "Author" used singular, while the manuscript has two authors.

Response: Thanks for your kind reminders. We have corrected this issue. [Pg309]

[Reviewer #3-Specific comment 8] Lines 283-287 "1-x", y, z and other symbols should be in italics "1-x", y, z

Response: Thanks for your kind reminders. We have corrected this issue. [Pg311-312, Pg486-488,Pg500-502,Pg517-]

[Reviewer #3-Specific comment 9] Line 291 FR should be subscript FR

Response: Thanks for your kind reminders. We have corrected this issue. [Pg301]

[Reviewer #3-Specific comment 10] Line 363-364 "parameters" ... "are many" - something is wrong with the grammar

Response: Thanks for your kind reminders. We have reorganized the sentence‘The parameters of the tripartite evolutionary game model of this system are many and complex, and only some of them are selected for analysis due to the limited space of the paper.’[Pg382-384]

[Reviewer #3-Specific comment 11] Line 390 "production SRW sector" should be "SRW production sector"

Response: Thanks for your kind reminders. We have corrected this issue.[Pg410]

Round 2

Reviewer 1 Report

Manuscript can be publoshed in present form.

Reviewer 2 Report

The reviewer appreciates the authors' efforts to improve the quality of the submitted article. After the revision, the manuscript, as a whole is more logically described and scientifically sound.